# Impact of Climate Change on Land Use, Yield and Production of Cassava in Thailand

**Siwabhorn Pipitpukdee** [1,2]**, Witsanu Attavanich** [1,]*** and Somskaow Bejranonda** [1]

[1]   Department of Economics, Kasetsart University, 50 Phahonyothin Rd., Chatuchak, Bangkok 10900, Thailand; siwabhorn.p@ku.th (S.P.); somskaow.b@ku.th (S.B.)
[2]   Center for Advanced Studies for Agriculture and Food, Kasetsart University Institute for Advanced Studies, Kasetsart University, Bangkok 10900, Thailand
*   Correspondence: witsanu.a@ku.ac.th

**Abstract:** This article examined the effect of climate change on land use, yield, and production of cassava in Thailand, employing the panel data analysis between 1989 and 2016. The spatial regression and the instrumental variable method with the generalized method of moment were employed to address the endogeneity problems with the unique climate dataset. The current article investigated that total rainfall and the La Niña event determined harvested area of cassava. In addition, the harvested area was decreased as the population density increased due to high demand for non-agricultural use. On the other hand, increased access to irrigation systems enhanced the harvested area of cassava. Considering the yield of cassava, we found an inverted U-shape relationship between yield and temperature. Moreover, this study revealed that climate variability, extreme events and technological progress statistically influenced cassava yields. By using the climate projections during 2046–2055, we found that harvested area and yield of cassava were projected to reduce 12.49–16.05% and 2.57–6.22% from the baseline. As a result, cassava production in Thailand was predicted to decline 14.74–21.26% from the baseline. The well-being of a half-million farmers in Thailand plus actors in the global supply chain of cassava will be vulnerable to climate change.

**Keywords:** climate change impacts; cassava; land use; yield; production; Thai agriculture

## 1. Introduction

Cassava is an important staple food for over 800 million people globally [1], providing a basic diet for people in several countries, especially in Sub-Saharan Africa with the highest supplier of carbohydrates among staple crops [2]. It is also an important energy crop for bioethanol production in several countries [3] and can grow mostly in the hotter low land tropics and also depleted soils [4]. Among cassava producing countries, Thailand is a main producer and exporter of cassava products. For the global perspective, Thailand has ranked first in terms of the export values for the last decade of fresh cassava and manioc starch, accounting for 62.32% and 72.31% of global export values in 2019, respectively [5]. In addition, Thailand has ranked second and third in terms of cassava production and harvested area, with 32 million tons and 1.38 million hectares in 2018, respectively [6]. For the local perspective, there are approximately 0.46 million farm households growing cassava, with a million hectares of harvested area, ranked fourth among crops in the country in terms of cropland use [7].

Previous studies found that crop production can be affected by several other factors, including climate change, that influence harvested area and crop yields with direct impact via increasing temperature, changing patterns of rainfall, or indirect impact via soil, nutrient and increasing pests damage [8–12]. Even though cassava is resilient to climate change due to its intra-seasonal drought and tolerance of high temperatures, a prolonged drought period can reduce a root yield up to 60% [13].

During the past two decades, studies have investigated climate change impact on cassava yield and founded heterogenous effects across regions in the world.

For example, Tatsumi and colleagues [14] obtained the observational data during 1990–1999 and showed that cassava yield will be dropped in Central America, Polynesia, Northern Africa, Western Africa, South Asia and Central Asia ranging from −0.65% to 29.35%, while it will be increased in Caribbean, South America, Eastern Asia and Middle Africa ranging from 2.02% to 10.49% under the Special Report on Emissions Scenarios (SRES) A1B scenario in the 2090s. However, Adhikari and colleagues [15] reviewed previous studies investigating the impact of climate change on cassava production in Africa and concluded that cassava yield in the future is projected to drop approximately 8% or increase 10% of its yield based on different projection scenarios. Using panel data from 37 countries in Sub-Saharan Africa during 1961–2002, Blanc [16] projected that cassava yield will almost be unchanged in 2100 under SRES scenarios (A1FI, A2, B1, B2) compared to the baseline.

A recent study considered the effect of climate change on cassava in Nigeria, the largest global producer, and found that increase in rainfall has a mixed effect on cassava yield, depending on the location (upland or coastal) [17]. The study revealed that increase in rainfall positively affected cassava yield in the upland area, but it negatively influenced cassava yield in the coastal area. In addition, the authors of Reference [18] found that rainfall did not affect cassava yield, while temperature negatively affected cassava yield in the humid forest agro-ecological zone of Nigeria. While a majority of previous studies investigated the impact of climate change on cassava in Africa, they only focused on cassava yield and did not incorporate the role of land use in the analysis. Several studies revealed that increasing populations have increased the pressure on agricultural land for non-agricultural use [9,19]. Therefore, increases in cassava yield cannot ensure the increase in its production, food and energy security in many countries.

In Thailand, there is surprisingly no study attempting to project the implications of climate change on cassava production at the national level and consider changes in yield and harvested land simultaneously, even though cassava production in Thailand has played a key role in Thailand's agriculture and the global market. Therefore, the objective of this study is to explore the impact of climate change on harvested area, yield and production of cassava in Thailand utilizing the provincial-level panel data from 1989 to 2016. We also aim to forecast the future changes in harvested area, yield and production of cassava under climate change scenarios (Representative Concentration Pathways (RCPs) 4.5 and 8.5) derived from the Intergovernmental Panel on Climate Change (IPCC)'s Fifth Assessment Report [12].

In the next section, we present detailed materials and methods used in the current study. Then, in Section 3, we show empirical results and provide a discussion in detail. Last but not least, we summarize all findings and propose policy recommendations in Section 4.

## 2. Materials and Methods

### 2.1. Method

Following previous literature [8,9], the current article set up the harvested area and yield models at the province level to estimate the impact of climate change on cassava production in Thailand. Equations (1) and (2) below show the harvested area and yield models, where $i$ and $t$ are indexed for province and year, respectively.

$$Area_{it} = \alpha_0 + \alpha_1 Climate_{it} + \alpha_2 Price_{it} + \alpha_3 Irrigation_{it} + \alpha_4 Popdensity_{it} + \alpha_5 T_{it} + \alpha_6 T_{it}^2 + v_i + e_{it} \quad (1)$$

$$Yield_{it} = \beta_0 + \beta_1 Climate_{it} + \beta_2 Price_{it} + \beta_3 Irrigation_{it} + \beta_4 T_{it} + \beta_5 T_{it}^2 + u_i + \epsilon_{it} \quad (2)$$

$Area_{it}$ and $Yield_{it}$ are dependent variables capturing the harvested area and yield of cassava in province $i$ at year $t$. From now on, explanation for the subscripts will be omitted for brevity. **Climate** is the vector of climate factors consisting of growing season temperature, extreme maximum temperature,

total rainfall, maximum rainfall in 24 h and the dummy variables capturing extreme events (i.e., El Niño, La Niña and neutral phases of El Niño–Southern Oscillation). The growing season of cassava in Thailand used in this study is between January and December, since cassava is usually harvested at the age of 8–14 months. The vector of prices (**Price**) is composed of farm-received price of cassava and labor wage rate. *Irrigation* is the percent of irrigated area to total area. $T$ and $T^2$ are time trend capturing the advance in production technology. *Popdensity* in Equation (1) captures population density that will have the implication on demand of agricultural land for non-agricultural use. Moreover, $\alpha$ and $\beta$ are vectors of parameters to be estimated. Because factors affecting cassava harvested area and its yield may be omitted due to data limitation (e.g., soil quality, altitude), we incorporate $u$ and $v$ as region fixed effects in the models to control for omitted variables. Finally, $e$ and $\epsilon$ are error terms.

To explore the impacts of climate factors and controlling for other factors influencing harvested area and yield of cassava, this article uses an instrumental variables method with the generalized method of moment (GMM) to address the endogeneity bias of using input and output prices in the model [8]. Following the method of selecting the appropriate instrumental variables suggested by References [9,20], we selected one-year lagged variables of mortgage policy, Southern Oscillation Index (SOI) and total amount of rainfall as instrumental variables (IVs) of that harvested area model. On the other hand, we employed one-year lagged variables of the Alternative Energy Development Plan (AEDP) policy aiming to improve the yield of cassava for ethanol production, total amount of rainfall and global cassava production as IVs of the cassava yield model. In addition, we used Moran's I error and Lagrange multiplier lag recommended by Reference [21] to test for the possible spatial autocorrelation because harvested area and yield of cassava in a selected province may be correlated with nearby provinces. We also applied the White test firstly introduced by White [22] to test for the problem of heteroscedasticity for the panel data. We detected both heteroscedasticity and spatial autocorrelation in the error and lagged dependent variable. To address these problems, this paper uses the Huber–White standard error and the spatial panel autoregressive regression in both yield and harvested area models. Lastly, we employed the random effect model for the analysis as a result of the Hausman test [20].

After the estimation, we then used estimated coefficients from harvested area and yield models to predict the effect of climate change on cassava's harvested area and yield in the future using climate change projections from the IPCC's Fifth Assessment Report [12], with two scenarios, including RCP 8.5 (the worst case scenario that captures a rising radiative forcing pathway leading to 8.5 W/m$^2$ in 2100) and RCP 4.5 (the best case scenario that is stabilization without overshoot pathway to 4.5 W/m$^2$ at stabilization after 2100). Finally, we multiplied the predicted harvested area to yield to obtain the predicted future supply of cassava.

## 2.2. Data

To investigate the impact of climate change on cassava production, this study collected the data from several sources at the provincial level over 77 provinces in Thailand from 1989 to 2016. We collected harvested area, yield and farm-received price of cassava from the Office of Agricultural Economics [23] and obtained irrigation area from the Royal Irrigation Department [24]. In addition, minimum wage rates of labor were gathered from the Ministry of Labor [25]. Climate variables for each climate station (i.e., mean temperature, extreme maximum temperature, total precipitation, maximum rainfall in 24 h) were collected from the Meteorological Department. We obtained 2046–2055 climate projections from the IPCC's Fifth Assessment Report [12]. We calculated three phases of El Niño-Southern Oscillation (ENSO) including El Niño, La Niña and Neutral using the Oceanic Nino Index provided by the National Oceanic and Atmospheric Administration (NOAA) [26]. To capture changes in demand of land for non-agricultural development, this study calculated the population density by obtaining the population and land area statistics from the Ministry of Interior [27] and used the estimated future population under the assumption of moderate fertility rate from the Office of the National Economic and Social Development Council (NESDC) [28].

Figure 1 illustrates the historical monthly deviation of climate variables from the mean value during the period of 1989–2016. Figure 1a,b shows the monthly deviation of mean temperature and extreme maximum temperature in degrees Celsius (°C), while Figure 1c,d demonstrates the monthly deviation of total precipitation and maximum precipitation in 24 h in millimeter. Overall, we can observe that the monthly deviation of mean and variability of climate variables was changing overtime from 1989 to 2016. The monthly deviation of mean temperature and extreme maximum temperature clearly increased during the last decade.

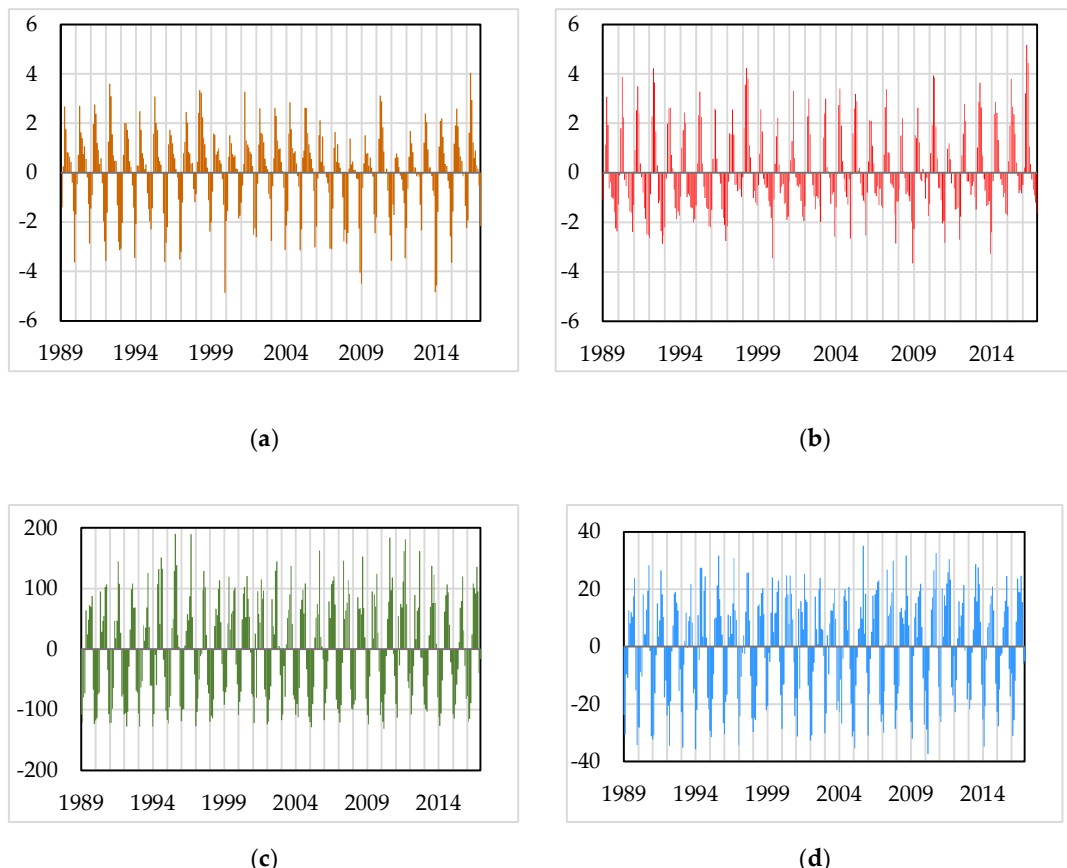

(**a**)　　　　　　　　　　　　　　　　　　　　　　　(**b**)

(**c**)　　　　　　　　　　　　　　　　　　　　　　　(**d**)

**Figure 1.** Historical monthly deviation of climate variables from the mean value during the period of 1989–2016. (**a**) Monthly deviation of temperature (Celsius). (**b**) Monthly deviation of extreme maximum temperature (Celsius). (**c**) Monthly deviation of precipitation (mm). (**d**) Monthly deviation of maximum precipitation in 24 h (mm).

While the climatic data are available from each climate station, the agricultural data are reported at the provincial level. Moreover, each climate station is not located at the center of the province. Also, some provinces have several climate stations and some of them have no climate station. In addition, there may be climate variation in provinces that have a large area. Following Reference [8], we addressed this mismatch problem of data by using a spatial statistical analysis to link the agricultural data and the climate data together. We used weighted least square regression to estimate provincial-average climates by controlling for the distance from the province centroid to weather station, latitude, longitude and height of weather station. Because closer stations generally provide more information about the centroid's climate, the weight used in the regression is the inverse of the square root of a station's distance from the province centroid. We located the centroid of each province by the red pin and constructed a 250 km radius circle by assuming that all the climate stations (yellow pins) within this radius provide some useful climate information (see Figure 2 for an example). In the next step, since the set of stations within 250 km and the weights (square root of a station's distance) are unique for

each province, we estimated a separate regression for each province. The regression fits a second-order polynomial over four climate variables plus a constant term. In total, 8624 regressions were estimated from 4 final variables for each of the 77 provinces and 28 years.

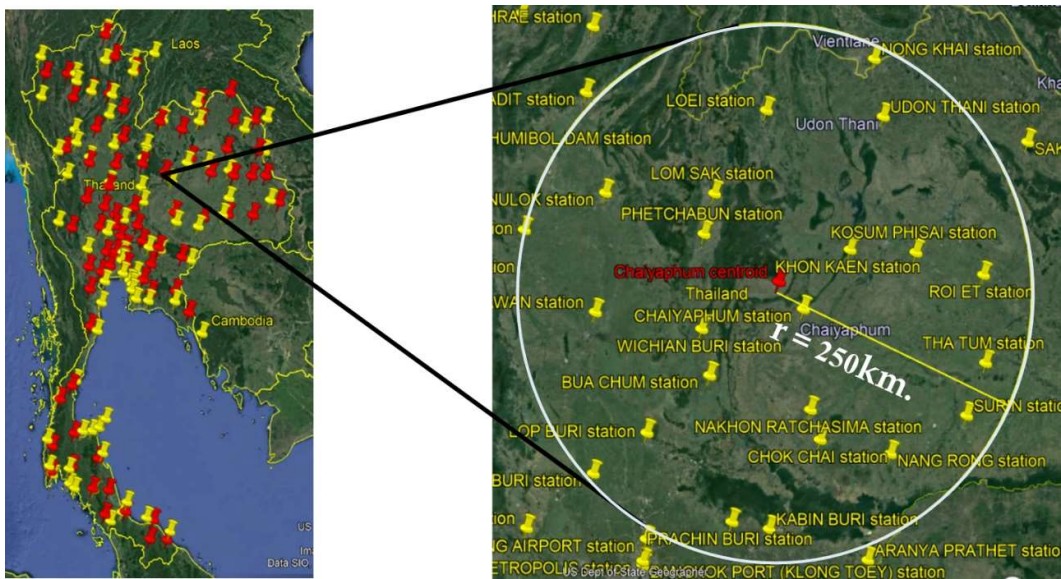

**Figure 2.** Weather stations within the radius of 250 km from the centroid of a selected province.

## 3. Results and Discussion

This section starts by providing the summary statistics of variables used in the models. Then, we show and discuss the estimated coefficients from harvested area and yield models. Finally, we reveal the predicted changes of cassava harvested area, yield and corresponding production under climate change scenarios, as well as discussing key findings. Table 1 reveals that the average harvested area of cassava from 1989 to 2016 is 26.74 thousand hectares per province, with the average yield of 17,440.85 kg per hectares. For the climate variables during the same period, the annual average temperature and the extreme maximum temperature are 27.46 and 35.92 °C, respectively. The maximum rainfall in 24 h and total annual rainfall are 33.87 mm per day and 1346.04 mm, respectively. Over the period of study, the population density is 112.62 persons per square meter. The one-year lag price of cassava and wage rate are 51.64 US dollars per ton and 6.42 US dollars per day, respectively. Lastly, the average irrigated area accounts for 7.82% of the total province area.

**Table 1.** Summary statistics of selected variables at the provincial level.

| Selected Variables | Mean | SD | Min | Max |
|---|---|---|---|---|
| Harvested area (1000 HA) | 26.74 | 41.42 | 0.04 | 317.40 |
| Yield (kg/HA) | 17,440.85 | 3445.39 | 10,443.75 | 27,550.00 |
| Average temperature (°C) | 27.46 | 0.72 | 25.03 | 29.09 |
| Maximum rainfall in 24 h (mm/day) | 33.87 | 4.00 | 22.98 | 47.28 |
| Extreme maximum temperature (°C) | 35.92 | 0.55 | 34.34 | 37.38 |
| Total rainfall (mm) | 1346.04 | 206.51 | 886.76 | 2007.98 |
| Population density (Person/sq.km) | 112.62 | 46.16 | 21.51 | 339.11 |
| Lag received price (USD/t) | 51.64 | 17.24 | 18.28 | 116.21 |
| Lag wage (USD) | 6.42 | 1.26 | 4.88 | 9.69 |
| % Irrigated area per province area | 7.82 | 10.95 | 0 | 60.68 |
| No. of observation | 1242 | | | |

Note: SD, Min and Max are abbreviations for standard deviation, minimum, and maximum.

## 3.1. Estimated Results

Tables 2 and 3 reveal the estimated coefficients from the harvested area and corresponding yield models of cassava for all provinces considered together using the provincial panel data. Details are explained in Sections 3.1.1 and 3.1.2, respectively.

**Table 2.** Factors influencing the harvested area of cassava.

| Variables | Coefficients | Standard Errors |
|---|---|---|
| Time trend | −2.92 *** | 0.92 |
| Time trend square | 0.12 *** | 0.04 |
| Population density | −0.09 *** | 0.03 |
| % Irrigated area per province area | 0.25 ** | 0.12 |
| Total rain | 0.02 | 0.03 |
| Total rain square | $-1.95 \times 10^{-5}$ * | $1.10 \times 10^{-5}$ |
| Maximum rain in 24 h | −0.44 | 0.43 |
| Extreme max temperature | −0.48 | 2.42 |
| El Niño | −3.80 | 2.54 |
| La Niña | 20.24 *** | 6.74 |
| North | 1.63 | 5.31 |
| Northeast | 42.57 *** | 5.5 |
| East | 33.35 *** | 6.11 |
| Lag price | −0.66 ** | 0.28 |
| Lag wage | 2.22 | 1.57 |
| Constant | 76.22 | 83.43 |
| Observations | 1242 | |
| R-square adjusted | 0.84 | |
| Root MSE | 10.58 | |

Notes: *, ** and *** indicate significance at the 10%, 5% and 1% level, respectively.

**Table 3.** Factors influencing cassava yield.

| Variables | Coefficients | Standard Errors |
|---|---|---|
| Time trend | 82.29 | 100.08 |
| Time trend square | 13.63 *** | 4.68 |
| % Irrigated area per province area | −21.85 *** | 7.18 |
| Average temperature | 18,965.98 *** | $4.95 \times 10^{3}$ |
| Average temperature square | −365.73 *** | 91.03 |
| Total rain | −6.47 *** | 2.29 |
| Total rain square | $5.79 \times 10^{-4}$ | $7.79 \times 10^{-4}$ |
| Maximum rain in 24 h | 78.03 *** | 27.22 |
| Extreme max temperature | 1438.41 *** | 270.30 |
| El Niño | −1495.48 *** | 209.04 |
| La Niña | 1198.04 ** | 585.61 |
| North | −2259.85 *** | 403.96 |
| Northeast | −1640.49 *** | 344.84 |
| East | 3210.47 *** | 379.92 |
| Lag price | −59.16 ** | 27.69 |
| Lag wage | −617.81 *** | 75.68 |
| Constant | $-2.71 \times 10^{5}$ *** | 68,636.46 |
| Observations | 1242 | |
| R-square adjusted | 0.75 | |
| Root mean square error (MSE) | 1487.28 | |

Notes: ** and *** indicate significance at the 5% and 1% level, respectively.

### 3.1.1. Factors Influencing the Harvested Area of Cassava

Considering climate variables shown in Table 2, we find that total rainfall and the La Niña phase determine harvested area of cassava. While the linear term of total rainfall is not statistically significant, its square term is statistically significant with small magnitude. This finding implies that the effect of rainfall on harvested area of cassava is small with an inverted U-shape relationship between rainfall and harvested area of cassava. Moreover, harvested area of cassava during the La Niña phase is higher than that in the neutral phase. In addition, increases in the percent of irrigated area to total provincial land area expanded harvested area of cassava as expected. For the change in the socio-economic condition, we observed that an increase in population density reduces the cassava harvested area due to increased demand of land for non-agricultural use.

Moreover, one-year lagged price negatively affects harvested area of cassava. Higher expected cassava price could incentivize farmers to switch a part of their land to other crops for risk diversification. The technological progress captured by the variable "Time trend" and its square term reveals a U-shape relationship between harvested area of cassava and the technological progress. In other words, the harvested area is reduced as technology is improved in the initial period, while it is increased as technology is improved in the later period. Considering the regional fixed effects, we observed that the harvested area of cassava in the Northeastern and Eastern regions is larger than that in the Central region, with statistical significance at the 1 percent level. The harvested area of cassava located in the Northern region is not statistically different from the Central region. This finding is consistent to the historical evidence that the Eastern region was first to be promoted to grow cassava by the government, followed by the Northeastern region.

### 3.1.2. Factors Influencing Cassava Yield

Table 3 reveals the inverted U-shape relationship between average temperature and cassava yield, as found in several previous studies [8,9]. The negative impact of high temperature on cassava yield is also revealed by other studies [29] and cassava yield is influenced by both mean temperature and total rainfall [30]. Total rainfall is negatively correlated to cassava yield, consistent with previous studies [16,31], while increases in extreme maximum temperature and maximum rainfall within 24 h enhance cassava yield. Because a majority of harvested areas of cassava in Thailand are located in dryland above the sea-level, therefore, the problem of flash flood from heavy rainfall is a minor issue. Instead, heavy rainfall within 24 h helps accumulate water and soil moisture, which is beneficial to cassava yield. Considering extreme events, we revealed that the cassava yield during the El Niño phase was lower than the yield during the neutral phase, while the cassava yield during the La Niña phase was greater than the yield during the neutral phase.

Our study also investigates that cassava yield is adversely affected by increases in percent of irrigated area to total land area, since the cassava root stays underground and the irrigated area generally locates in the lowland area, which is risky to flood that can damage cassava yield. Moreover, it is popular in Thailand to plant rice in the irrigated area. Farmers usually flood the water into the rice field, making it difficult for nearby farms to plant cassava. Furthermore, one-year lag farm-received price and labor wage rate are inversely related to cassava yield, with statistical significance at 5 and 1 percent, respectively. The technological progress is found to enhance cassava yields. Finally, the cassava yields in the Northern and Northeastern regions generally located in the upland area are lower than the cassava yield in the Central region. On the other hand, cassava yield in the Eastern region located near the coastal area is higher than cassava yield in the Central region.

### 3.2. Projected Impact of Climate Change on Harvested Area, Yield and Production of Cassava

This subsection predicts the impact of climate change on cassava production by comparing the future period (2046–2055) to the baseline period (1992–2016). All climate projections were obtained

from IPCC AR5 [12]. We then investigated the ranges of future impact of climate change under representative concentration pathways (RCPs) 4.5 and 8.5.

Table 4 presents the regional projected changes in growing season temperature and annual maximum precipitation within 24 h during 2046–2055 under RCPs 4.5 and 8.5 from the baseline during 1992–2016. Among regions, the Northeastern region is predicted to have the highest increase in growing season temperature and extreme maximum temperature from the baseline, ranging from 1.22–1.68 °C and 1.55–1.86 °C under RCPs 4.5 and 8.5, respectively. Annual maximum precipitation within 24 h of regions is predicted to increase, ranging from 0.04 to 0.07 mm and 0.11 to 0.19 mm under RCP 4.5 and RCP 8.5, respectively.

**Table 4.** Projected changes in growing season temperature (°C) and annual maximum precipitation within 24 h (mm) during 2046–2055 under representative concentration pathways (RCPs) 4.5 and 8.5 from the baseline during 1992–2016.

| | Projected Changes in Growing Season Temperature (°C) | | Projected Changes in Average Annual Extreme Maximum Temperature (°C) | | Projected Changes in Annual Maximum Precipitation Within 24 h (mm) | |
|---|---|---|---|---|---|---|
| | RCP 4.5 | RCP 8.5 | RCP 4.5 | RCP 8.5 | RCP 4.5 | RCP 8.5 |
| Central | 1.08 | 1.48 | 1.21 | 1.61 | 0.07 | 0.12 |
| North | 1.22 | 1.65 | 1.52 | 1.80 | 0.06 | 0.19 |
| Northeast | 1.22 | 1.68 | 1.55 | 1.86 | 0.04 | 0.11 |
| East | 1.08 | 1.48 | 1.21 | 1.61 | 0.07 | 0.12 |

To better understand the heterogenous effects of total annual rainfall at the local level, we used the downscaled IPCC AR5 climate projections at the watershed level. Currently, there are 25 watersheds in Thailand. Overall, we observed that the total amount of annual rainfall under RCP 8.5 will be higher than the total amount of annual rainfall under RCP 4.5 (Figure 3) and it will be increased in all watershed areas (except for the lower-southern region) from the baseline.

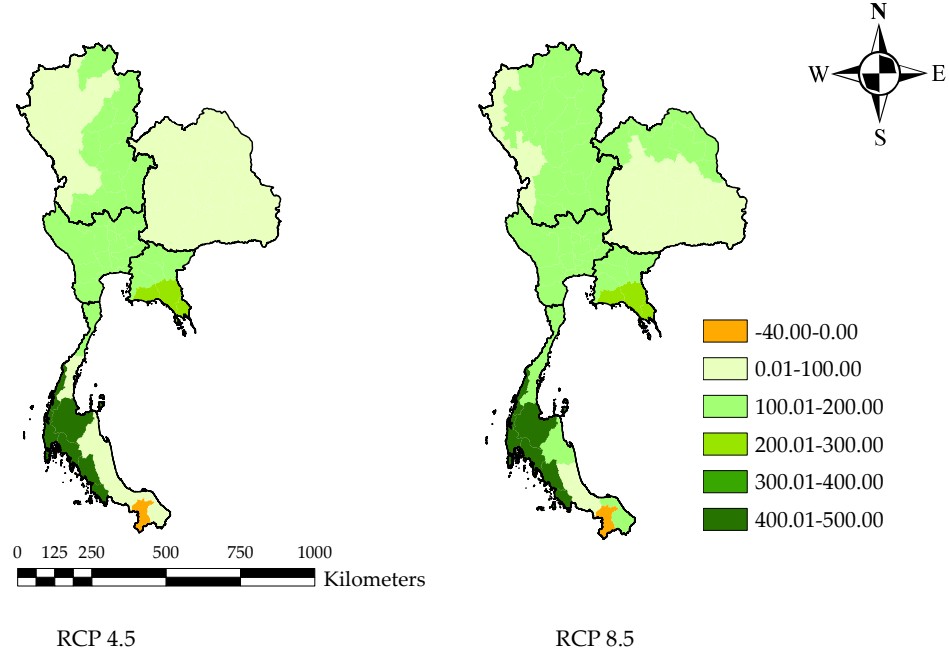

**Figure 3.** Projected changes in total annual rainfall (mm) during 2046–2055 under RCPs 4.5 and 8.5 from baseline during 1992–2016.

By applying the similar method to Reference [8], Figure 4b illustrates projected percent changes in the population density under the scenario of moderate fertility rate from the baseline (Figure 4a) during 2046–2055. We reveal that the population density is projected to decline in the Northeastern and Northern regions of Thailand. On the other hand, it tends to increase in the Central, Eastern and Southern regions.

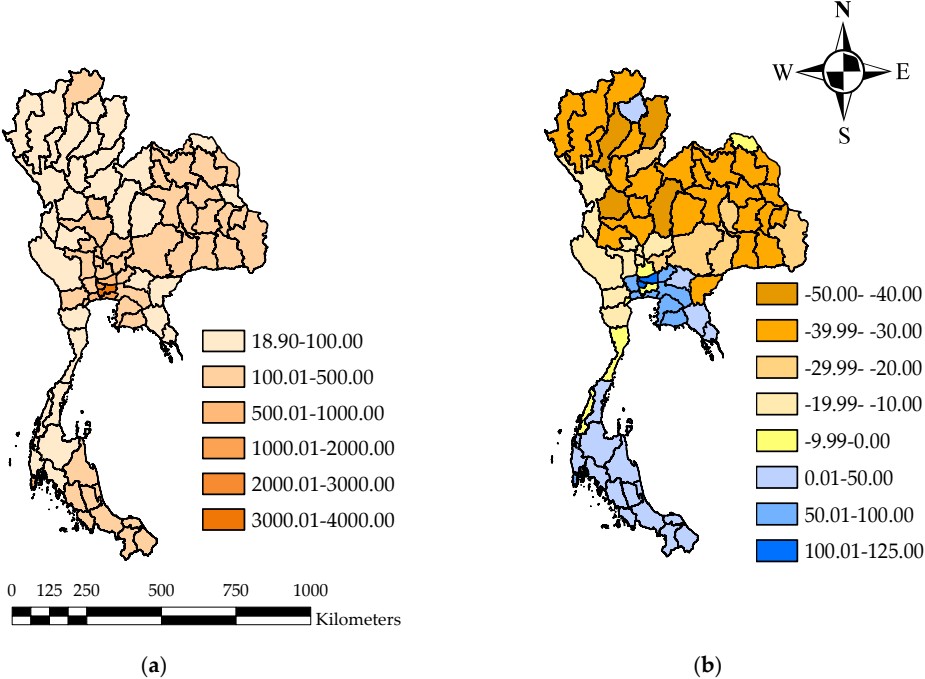

**Figure 4.** Projected changes in population density during 2046–2055 under scenario of moderate fertility rate from the baseline. (**a**) baseline of average population density during 1992–2016 (people/km$^2$), and (**b**) percent change of population density in 2046–2055 from the baseline.

In the next step, we incorporate projections of climate and population density in the estimated harvested area and yield models in Tables 2 and 3 to investigate the future changes in harvested area, yield and production of cassava. Here, we reveal that harvested area, yield and production of cassava are forecasted to drop in every scenario at the country level, as shown in Table 5. The harvested area of cassava is predicted to drop by 12.49% under RCP 4.5 and 16.05% under RCP 8.5 from the baseline. Future cassava yield is also projected to slightly decline by 2.57% under RCP 4.5 and 6.22% under RCP 8.5 from the baseline, consistent to findings from previous studies that reveal the negative effect of climate change on cassava yield [17].

**Table 5.** Projected changes in cassava production under RCPs 4.5 and 8.5 during 2046–2055 from baseline 1992–2016 at the national level.

| Variables | Baseline | Percent of Change under RCP 4.5 | Percent of Change under RCP 8.5 |
|---|---|---|---|
| Harvested area | 1212 (1000 ha) | −12.49 | −16.05 |
| Yield | 18,400 (kg/ha) | −2.57 | −6.22 |
| Production | 22.32 (1000 MT) | −14.74 | −21.26 |

Note: ha and MT are abbreviations for hectare and metric ton.

Moreover, by multiplying harvested area and corresponding yield of cassava, this study reveals that cassava production is predicted to decrease by 14.74% under RCP 4.5 and 21.26% under RCP 8.5 from the baseline, respectively. The reduction of cassava production will significantly affect global cassava traded in the world market, since Thailand's fresh cassava and manioc starch contributed 62.32% and 72.31% of global export values in 2019.

Next, this study investigates the heterogenous effects of climate change on cassava at the provincial level by incorporating the role of land use pressure captured by population density. Figure 5 reveals that the harvested area of cassava is predicted to decrease 33 out of 46 provinces ranging from 0.01% to 87.76% under RCP 4.5 and 40 out of 46 provinces ranging from 0.27% to 93.47% under RCP 8.5 from the baseline. On the other hand, a small number of provinces located in the Northeastern region (i.e., Surin, Si Sa Ket, Maha Sarakham, Buri Ram, Khon Kaen and Nong Bua Lam Phu) are projected to be faced with positive impacts of climate change under climate scenarios.

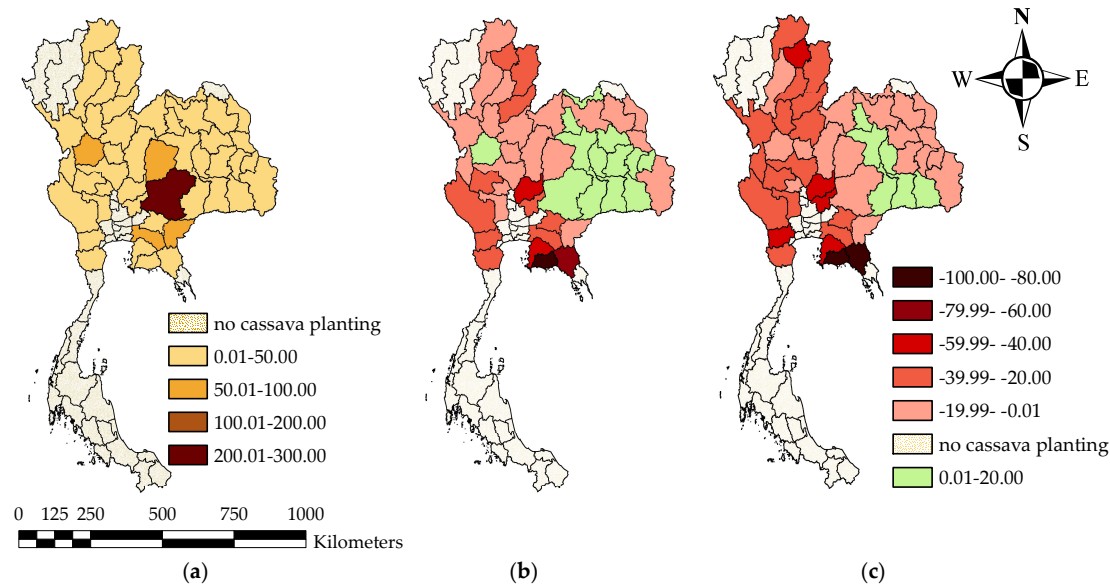

**Figure 5.** Projected percent changes in harvested area of cassava under climate change scenarios. (**a**) baseline harvested area (1000 ha), (**b**) percent of change in harvested area under RCP 4.5 and (**c**) percent of change in harvested area under RCP 8.5.

For the future change in cassava yield, this study reveals that 23 out of 46 provinces are projected to experience the reduction in cassava yield ranging from 0.41% to 9.01% under RCP 4.5, while 40 out of 46 provinces are predicted to encounter the decrease in cassava yield ranging from 0.15% to 12.20% under RCP 8.5 (Figure 6). Cassava yield in the Northern region is projected to increase under RCP 8.5, while it is forecasted to drop in other regions, with the largest drop in the Eastern, lower section of Central and Northeastern regions, respectively. Overall, cassava yield in the coastal and lowland areas receives higher, negative impacts from climate change than that in the upland area. These findings are similar to Isaiah and colleagues [17], who reveal that climate variables determine cassava yield differently in Nigeria depending on whether the farm is located in the upland or coastal region.

Finally, by multiplying harvested area and yield, Figure 7 reveals that, under RCP 4.5, the cassava production in the Northeastern region is forecasted to increase by 0.55–6.74%, while its production in other regions is projected to decline. We also observe that the cassava production is projected to drop in all provinces (excepting for Nong Kai), ranging from 0.14% to 94.27% under RCP 8.5. The top five provinces having the largest cassava production (i.e., Nakhon Ratchasima, Kamphaeng Phet, Chaiyaphum, Sa Kaeo and Chachoengsao), accounting for 43.05% of total cassava production, are projected to experience a reduction of 1.44–39.29% under RCP 4.5 and 8.73–43.72% under RCP 8.5 from the baseline.

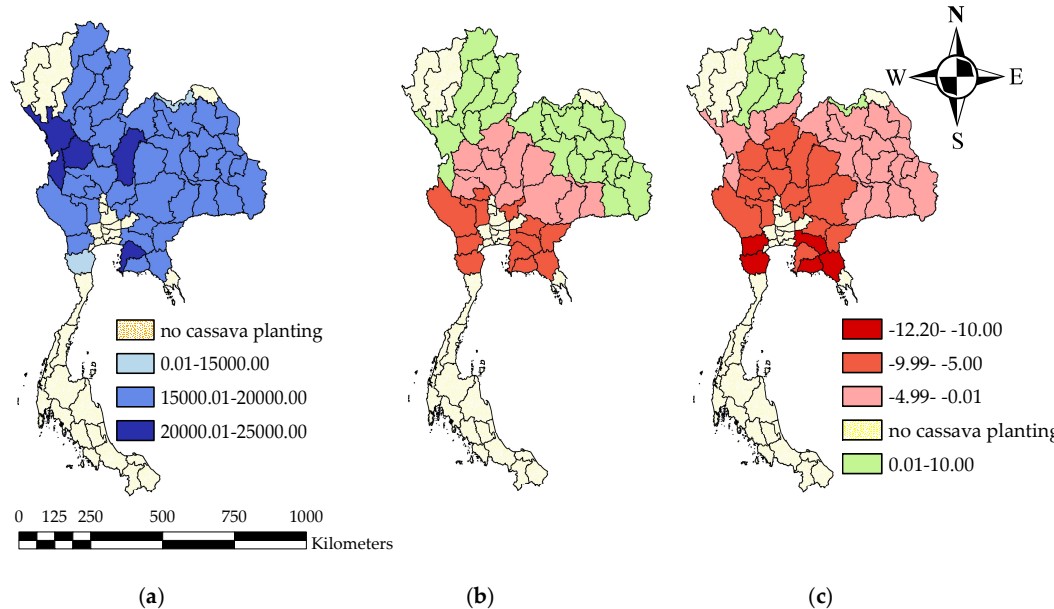

**Figure 6.** Projected percent changes in yield of cassava under climate change scenarios. (**a**) baseline yield (kg/ha), (**b**) percent of change in yield under RCP 4.5 and (**c**) percent of change in yield under RCP 8.5.

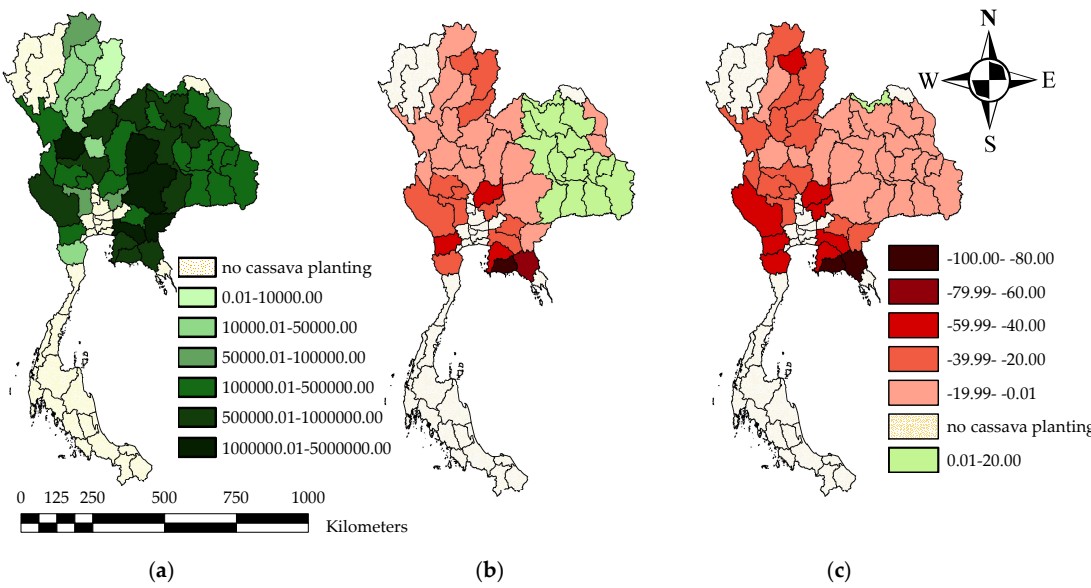

**Figure 7.** Projected percent changes in cassava production under climate change scenarios. (**a**) baseline production (MT), (**b**) percent of change in production under RCP 4.5 and (**c**) percent of change in production under RCP 8.5.

## 4. Conclusions

This study aimed to project the climate change impacts on cassava production derived from changes in harvested area and yield of cassava in Thailand, a major global producer and exporter of cassava products, using the instrumental variable method with the generalized method of moment and spatial regression. A panel data at the provincial level was constructed from 1989 to 2016 with the downscaled climate projections under RCP 4.5 and RCP 8.5 from IPCC AR5 and predicted future population density during 2046–2055.

Considering factors influencing harvested area of cassava, this study revealed that total rainfall and the La Niña phase determine cassava harvested area. The total rainfall is non-linearly determined the

harvested area with an inverted U-shape relationship. Besides climate variables, increasing population density also reduces the harvested area for non-agricultural use and the technological progress has a U-shape relationship with the harvested area of cassava. Considering factors influencing the cassava yield, we investigated that both the mean and variability of climate variables statistically determine cassava yield. Technological progress also statistically influences cassava yield with a non-linear effect. Moreover, farm-received price and labor wage rate in the previous year also affect cassava production.

For the projected impact of climate change, the current article investigated that the harvested area of cassava in 2046–2055 is projected to drop by approximately 12.49–16.05% from the baseline 1992–2016, while the harvested area is predicted to increase in some provinces located in the Northeastern region. The future cassava yield is projected to drop by 2.57–6.22% countrywide from the baseline. The cassava yield tends to increase in the upper section of the country, while it is projected to drop in the lower section of the country. By multiplying yield and corresponding harvested area, this study estimated that the countrywide cassava production is predicted to decrease by 14.74–21.26% from the baseline. In the world, with small change in climate conditions (RCP 4.5), a majority of provinces located in the Northeastern region are predicted to experience an increase in cassava production. Conversely, under the large change in climate conditions (RCP 8.5), all provinces (except Nong Khai province) are forecasted to encounter a drop in cassava production. These findings imply a significant decline in the amount of Thailand's export of cassava products to the world market.

Policymakers should raise awareness about the severity of climate change impacts on cassava production to farmers, farmer institutions and the private sector that uses cassava as a raw material. At the same time, it is recommended to provide the knowledge to farmers for the appropriate autonomous adaptation options, such as diversification of crop mix, moisture management, soil and water conservation and changing planting dates. According to the findings, planting cassava as an adaptation option to reduce impact of climate change in Thailand may not be a good choice if there is no assistance from the government in the form of public adaptation. For example, the government should expand irrigation infrastructure to affected areas as much as possible since almost all of the planted areas of cassava are located in the non-irrigated areas. Not only will this investment reduce the negative impact of climate change and improve the efficient use of farmland for cassava, it will also be beneficial to other farmers who plant other crops.

The priority of assistance should be firstly set in the high vulnerability areas such as Lopburi, Chon Buri, Kanchanaburi, Chachoengsao, Uthai Thani, Prachin Buri, Phitsanulok, Nakhon Sawan, Sa Kaeo and Tak, if the budget is limited. A crop insurance program for cassava and other crops may be needed to reduce the risk of farmers from climate change and its variability. Moreover, additional agricultural research funding may be needed to develop a new drought-tolerant cultivar and explore the optimal adaptation strategies. Over the past decade, the overall agricultural research expenditure in Thailand has been diminished, and past research studies focusing on issues related to climate change adaptation options are limited for cassava. Finally, besides farmers, actors across the supply chain of cassava products, especially China, a major global importer of cassava, as well as traders of cassava-related products, should include the impact of climate change to their future planning.

**Author Contributions:** Conceptualization, W.A. and S.B.; methodology, S.P., W.A., and S.B.; formal analysis, S.P. and W.A.; investigation, W.A.; writing—original draft preparation, S.P.; writing—review and editing, W.A.; visualization, S.P.; supervision, W.A. All authors have read and agreed to the published version of the manuscript.

**Funding:** This research was partially funded by the Center for Advanced Studies for Agriculture and Food, Institute for Advanced Studies, Kasetsart University, under the Higher Education Research Promotion and National Research University Project of Thailand, Office of the Higher Education Commission, Ministry of Education, Thailand.

**Conflicts of Interest:** The authors declare no conflict of interest.

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
