# Peer review of "Impact of Climate Change on Land Use, Yield and Production of Cassava in Thailand"

_agriculture, doi:10.3390/agriculture10090402_

Round 1
Reviewer 1 Report
The Introduction section and the case made for the relevance of the study, as well as the use of a large database merging province-level data with IPCS models, are commendable characteristics of this manuscript.
The Results and Discussion section need some improvement:
- There are claims in the Results and Discussion section that are not fully apparent by the Tables included in the manuscript. For example under 3.1.1 Factors influencing the harvested area of cassava it was claimed that "total rainfall and the La Niña determined cassava harvested area. " but the coefficient of total rainfall is small and non-significant.
- Tables 2 and 3; is not clear if they refer to all provinces considered together or to a specific one. Please clarify this. In the manuscript you indicate significant differences in climate variables between the coastal and upland regions in Nigeria, why these distinctions are not address here?
- Introduce int he text the variables; North, Northeast, and East, present in Tables 2 and 3.
- It is claimed that "technological progress non-linearly influenced cassava harvested area with U-shape relationship. " The coefficient to the linear progress variable is larger (and negative) than the the coefficient of the non-linear variable. Why discuss only the non-linear one?
- this sentence "The total rainfall was non-linearly determined cassava 169 harvested area with inverted U-shape relationship. " Is not clear.
- Because you are presenting your results here, please use the present tense of the verbs; e.g., use "are revealed" instead of "were revealed".
- Northeast and East coefficient are significant and large in the model for the harvest area of cassava; similarly for North, Northeast, and East for the yield. Why they are not discussed?
Following these comments you might want to go back to the Conclusion section and expand the evidence for your conclusions here as well.
Reviewer 2 Report
Comments
SUMMARY
The paper addresses the research area related to “Impact of climate change on agriculture” of the MDPI Agriculture journal. I believe that the target journal is an appropriate forum for this article. The paper aims to estimate the impact of climate change on harvested area, yield, and production of cassava in Thailand using the provincial-level panel data. Then, they project future changes in harvested area, yield, and production of cassava under climate change scenarios (RCPs 4.5 & 8.5) derived from the Intergovernmental Panel on Climate Change (IPCC)’s Fifth Assessment Report.
BROAD COMMENTS
This study is of great importance for Thai agriculture and economy. The Introduction section is well done. The methodology is well written and detailed. The results are well presented. However, the only weakness of this study is that the authors failed to discuss in-depth the findings presented in the results and discussion section. Besides, in the results and discussion section, the authors used the same references they have cited in the introduction section.
SPECIFIC COMMENTS
Figures 2-6: I suggest the authors add the north arrow and the scale bar to the maps.
Lines 79-114: Since the authors used the estimated coefficients α and β for predicting the impact of climate change on cassava yield and cultivated areas, they should conduct the heteroscedasticity test on the data before doing so to avoid bias. Also, the autocorrelation test is worth conducting in this study before the regression analysis.
Lines 115-149: The study is about the impact of climate change on cassava production and the authors used the dataset (1989-2016) including rainfall and temperature. However, we don’t have any idea about the annual variability and the trends of the rainfall and temperature during that period. So, I suggest the authors include in the manuscript some Figures and/or graphics about that. This will help the reader to understand easily the discussion of the findings.
Lines 150-272: The authors discussed the results of this study using the same studies (references) that they mentioned in the Introduction section and even in the method section. I suggest the authors discussed in depth the results of their study using additional references to the ones mentioned in the introduction section.
Lines 303-306: In the conclusion section, the authors referred to previous studies [27]. This makes this part of the conclusion looks like a literature review. It should not be the case. So, I suggest the authors revise this part and remove the reference.
Round 2
Reviewer 2 Report
I have undertaken a review of the manuscript (revised) as well as the attached author responses to the initial review where I recommended major revisions. I am satisfied with the revisions made by the authors as they have addressed most, if not all, of my initial comments.
This manuscript is a resubmission of an earlier submission. The following is a list of the peer review reports and author responses from that submission.